# Three-dimensional X-ray diffraction imaging of dislocations in polycrystalline metals under tensile loading

Mathew J. Cherukara [1,2], Reeju Pokharel [3], Timothy S. O'Leary[4], J. Kevin Baldwin[4], Evan Maxey[1], Wonsuk Cha [1], Jorg Maser[1], Ross J. Harder[1], Saryu J. Fensin[3] & Richard L. Sandberg[4]

The nucleation and propagation of dislocations is an ubiquitous process that accompanies the plastic deformation of materials. Consequently, following the first visualization of dislocations over 50 years ago with the advent of the first transmission electron microscopes, significant effort has been invested in tailoring material response through defect engineering and control. To accomplish this more effectively, the ability to identify and characterize defect structure and strain following external stimulus is vital. Here, using X-ray Bragg coherent diffraction imaging, we describe the first direct 3D X-ray imaging of the strain field surrounding a line defect within a grain of free-standing nanocrystalline material following tensile loading. By integrating the observed 3D structure into an atomistic model, we show that the measured strain field corresponds to a screw dislocation.

[1] Advanced Photon Source, Argonne National Laboratory, Argonne, IL 60439, USA. [2] Center for Nanoscale Materials, Argonne National Laboratory, Argonne, IL 60439, USA. [3] Materials Science and Technology Division, Los Alamos National Laboratory, Los Alamos, NM 87545, USA. [4] Laboratory for Ultrafast Materials and Optical Science, Center for Integrated Nanotechnologies, Los Alamos National Laboratory, Los Alamos, NM 87545, USA. Correspondence and requests for materials should be addressed to M.J.C. (email: mcherukara@aps.anl.gov) or to S.J.F. (email: saryuj@lanl.gov) or to R.L.S. (email: sandberg@lanl.gov)

Material defects and the strain fields accompanying them influence a wide variety of material properties, from mechanical strength and failure, the efficiency of electronic devices[1], radiation resistance, battery performance[2] and crystal growth, and dissolution[3]. Consequently, the ability to control the types and distributions of various defect structures would enable the design of materials with tailored properties. Such defect engineering has been used to increase the mechanical strength of materials while retaining their creep strength[4], engineer topological deformation modes[5], build more efficient thermoelectrical devices[6,7], and improve the performance of energetic materials[8,9]. The ability to visualize dislocations and their long-range strain fields then, is vital to the improved design of functional and structural materials with pre-engineered defects. Transmission electron microscopy (TEM) has provided a means of imaging dislocation structures for several decades, but has been limited in that it provides a two-dimensional (2D) projection of a three-dimensional (3D) object and does not typically provide strain information. Recently 3D electron imaging using high-angle annular dark-field scanning TEM (ADF-STEM) tomography has been used to image dislocations in 3D[10], and additional techniques have been developed to quantify strain fields from dislocations in 2D[11]. ADF-TEM has also been used to image screw-dislocations side-on[12]. Electron microscopy techniques that can simultaneously image structure and strain in 3D, however, remains elusive. Furthermore, TEM requires sample thicknesses of ~100 nm, the preparation of which can lead to the development of additional strain due to processing of the sample.

As a complement to electron microscopy techniques, X-ray coherent diffraction imaging (CDI) can provide full 3D, local structural and strain information with ~10 nm resolution in structures ranging from nanoparticles[13–17] to foams[18] to biological structures[19,20]. Bragg CDI (BCDI) is a technique where 3D electron density and strains fields in a sample are obtained from scattered coherent X-rays in the far-field around a coherently scattered Bragg peak[19]. Iterative algorithms are then used to recover both the real space structure and "lost" phase information[21]. BCDI has been previously used to image line defect dynamics in battery nanoparticles during charge cycling[22], and to study the role of defects during crystal growth and dissolution[3]. However, with two exceptions[23,24], BCDI has been restricted to isolated nanoparticles (~0.5 μm in size) or extended single crystals through ptychography[25,26]. Both of the previous BCDI studies on polycrystalline samples were performed on substrate supported thin films, which are not easily amenable to in situ characterization under mechanical loading.

In this work, we use BCDI to measure the strain state of an individual grain of copper (Cu) for the first time in a free-standing polycrystalline film following tensile loading. To better understand the complex amplitude and strain variation in the crystal that was observed, we imported the reconstructed 3D electron density of the grain into an atomistic model.

## Results

**Polycrystalline BCDI experiment.** Figure 1 shows a schematic of the experimental setup for BCDI performed at the coherent X-ray scattering beamline (34-IDC) at the advanced photon source (APS). Polycrystalline Cu film was placed on a tension rig in the center of a diffractometer. The photon energy of the X-ray pulses was set to 9.0 keV using a Si (111) monochromator. Diffracted X-ray pulses were collected by an ASI Timepix detector in the (111) Bragg geometry. By employing iterative phase retrieval algorithms[27,28], both the complex electron density $\rho(r)$ and the phase information $\phi(r)$ are recovered[29]. In turn, the phase information yields the atomic displacement field in the entire crystal volume through the relation $\phi(r) = \vec{Q}.\vec{u}(r)$, where $\vec{u}(r)$ is the atomic displacement and $\vec{Q}$ is the scattering vector (along 111). Hence, the projected lattice displacement $\left(u_{\vec{Q}}\right)$ along the scattering vector can be obtained. Additionally, from $u_{\vec{Q}}$, the projected strain in the lattice along the $\vec{Q}$ direction can be calculated from the gradient of the displacement field: $\epsilon_{\vec{Q}} = \nabla\left(\vec{u} \cdot \vec{Q}\right)\vec{Q}$.

**Thin-film sample synthesis.** Polycrystalline Cu thin-film samples were prepared through DC magnetron sputtering in a vacuum chamber with a base pressure of 5.0E-8 torr. The process pressure was 3 millitorr with 30 SCCM of Ar and a target power of 300 watts. The deposition rate was 0.6 nm/s and the substrates used were MgO single crystals with (111) orientation. The Cu thin films (≈2 μm thick) were subsequently removed from the MgO substrate through sonication. Figure 2 shows the grain orientation and size distributions of an as-deposited sample before tensile loading. Pole figures in Fig. 2b show the crystallographic textures. From the pole figures, we can see that the majority of the grains with their crystal plane normal are oriented along the (111) direction, yielding a moderate texture of ≈4 multiples of random density. Grain size distribution is shown in Fig. 2c, where a significant portion of the grains are between 500 and 700 nm in size, which is ideal for the X-ray spot size of 700 nm by 700 nm.

**BCDI characterization of plastic deformation.** A thin-film of Cu was placed on a tensioning rig at the center of the diffractometer and was illuminated by a coherent beam focused down to ~700 nm X 700 nm, with the detector placed on the {111} powder ring. Of the observed {111} Bragg spots resulting from the

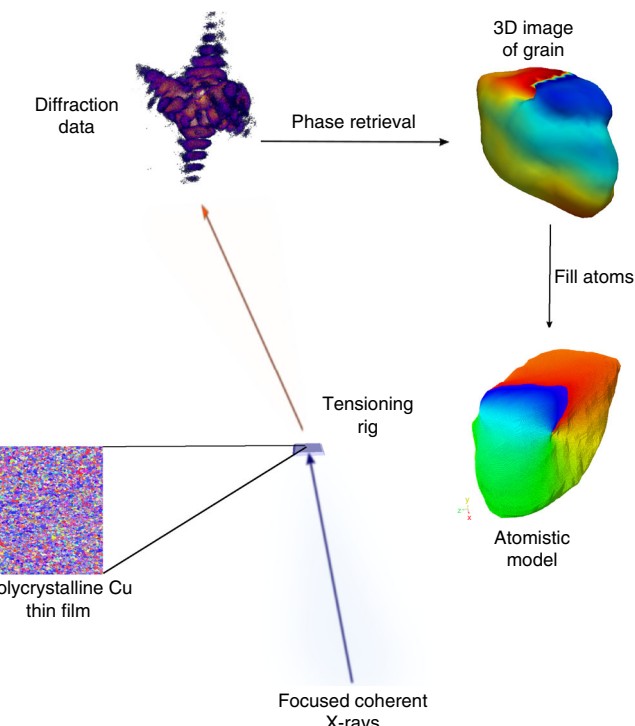

**Fig. 1** Coherent X-ray diffraction imaging in Bragg geometry. Focused coherent X-ray pulses are incident on the polycrystalline Cu sample. Diffracted X-ray pulses are recorded in the far-field by an ASI Timepix detector. The three-dimensional speckle pattern at the (111) Bragg peak is recorded from 2-D diffraction slices obtained by rocking the sample stage through small angles (~1⁰)

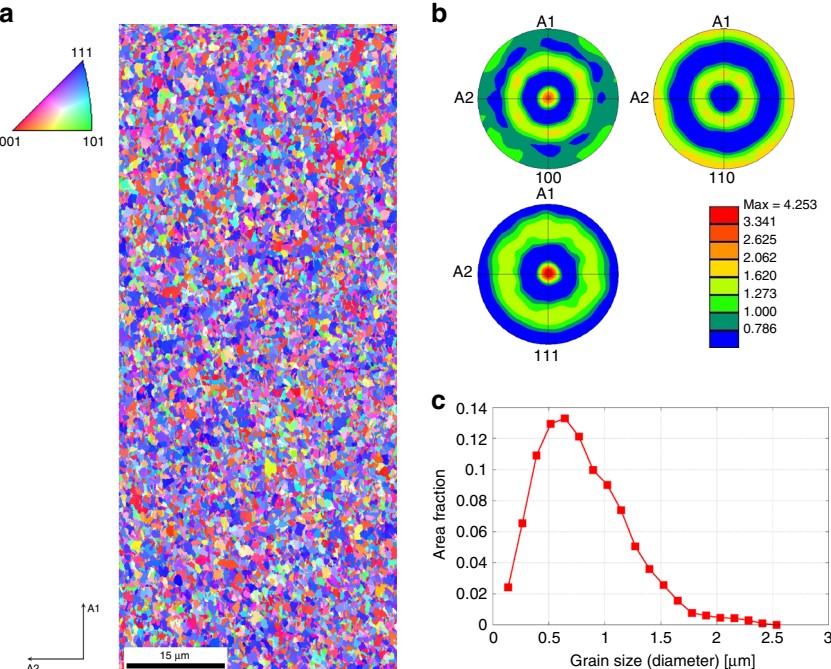

**Fig. 2** Characterization of Cu thin films. **a** EBSD micrograph showing the inverse Pole Figure map of the as-deposited Cu thin-film. **b** Pole figures and **c** grain size distribution

illumination of several grains in the sample, we focused on one that was relatively isolated on the powder ring to eliminate interference from other similarly oriented grains. The sample was loaded in tension until the {111} Bragg peak shifted under the applied strain. We did not measure the strain-rate (although it is probably quasistatic), and the total amount of applied strain, but rather focused on the changes induced in the diffraction pattern under an applied load. To obtain the 3D coherent diffraction pattern of the chosen grain about its (111) Bragg peak, we rotated the sample $\theta$ stage through $0.6°$ in steps of $0.005°$ (120 2D slices). The 2D diffraction intensities recorded at each step were stacked in a 3D array to obtain the final diffraction image[30]. From the 3D diffraction pattern, we obtained the real space amplitude and phase information using iterative reconstruction algorithms as discussed previously (also see Methods). Figure 3a shows the reconstructed amplitude of an individual Cu grain. Figure 3b shows line-outs of the reconstructed amplitude across the surface of the reconstructed grain, from which we estimate the image resolution to be ~30 nm in X, ~50 nm in Y, and ~40 nm in Z[31]. Unusually, we observe a volume of low reconstructed intensity that extends through the thickness of the grain, suggesting the presence of a line defect[3]. This region of low intensity is encircled by white dashes in Fig. 3a. The corresponding phase or displacement map (Fig. 3c) shows a circular wrap around this extended defect, while the projected strain along the Q (Fig. 3d) shows a sharp discontinuity along a plane normal to the axis of the defect. A contour level of 0.3 in amplitude was chosen in the representations of lattice displacement (Fig. 3c) and strain (Fig. 3d).

**Experimentally informed molecular dynamics simulations**. To better understand the origin of this amplitude and strain variation, we built an atomistic model with an initial structure similar to the final one obtained from experiments. To accomplish this, an electron density threshold of 0.1 in amplitude was chosen to define the isosurface of the grain. The resulting volume was scaled down by a factor of 5 in every dimension and filled with FCC Cu

atoms so that the (111) crystallographic direction is oriented along the experimentally measured Q vector. The resulting structure has 5,521,267 atoms. We used the nanoSCULPT code, which uses the point-in-polygon algorithm to fill atoms into a given volume in the chosen coordinate frame[32]. Figure 4a shows the atomistic structure obtained by atom-filling the experimentally imaged grain. To test the hypothesis that the experimentally observed amplitude and strain variation is due to the presence of a line defect (in particular a screw dislocation), we displaced the atoms in the model from their idealized lattice positions as predicted by classical theory. We assumed the Burger's vector is along the (110) direction (associated with the full dislocation), which is the expected slip direction in an FCC lattice. The atoms are then displaced by an amount calculated by $u_{(110)}(\theta) = b\theta/2\pi$. Since the experimental measurement is only sensitive to the strain along the scattering vector, we project the lattice displacements on to the (111) direction as shown in Fig. 4b[33]. A comparison between the calculated atomic displacements without any atomic relaxation as shown in Fig. 4b, and the experimentally observed lattice displacements shown in Fig. 4d suggests that the experimentally observed phase structure is a result of the strain field from a screw dislocation[3,34]. Figure 4e–g show histograms of the projected lattice displacement corresponding to Fig. 4b–d. We note that the volume of low amplitude around the dislocation core (~20 nm) appears larger than expected (Fig. 3c), but is expected given the resolution of the reconstruction is (30–50 nm).

**Discussion**

Finally, we explore the energetics of the imaged dislocation structure. In particular, we examine the possible preference for a perfect dislocation as opposed to splitting into partials. To this end, we minimize the potential energy of the atom structure with a perfect screw dislocation using the conjugate-gradient method as implemented in LAMMPS (see Methods) and compare the resulting displacement field (https://www.lammps.sandia.gov[35]). Figure 4 shows the defect structures colored by their displacement fields projected along the (111) crystallographic direction before

energy minimization (Fig. 4a, b) and following minimization (Fig. 4c). We note that further energy minimization to a higher tolerance in forces and energies leads to the dislocation escaping to the free surface of the grain. A comparison between the experimentally imaged displacement field (Fig. 4d) and atomistic simulations suggests that the screw dislocation might not prefer

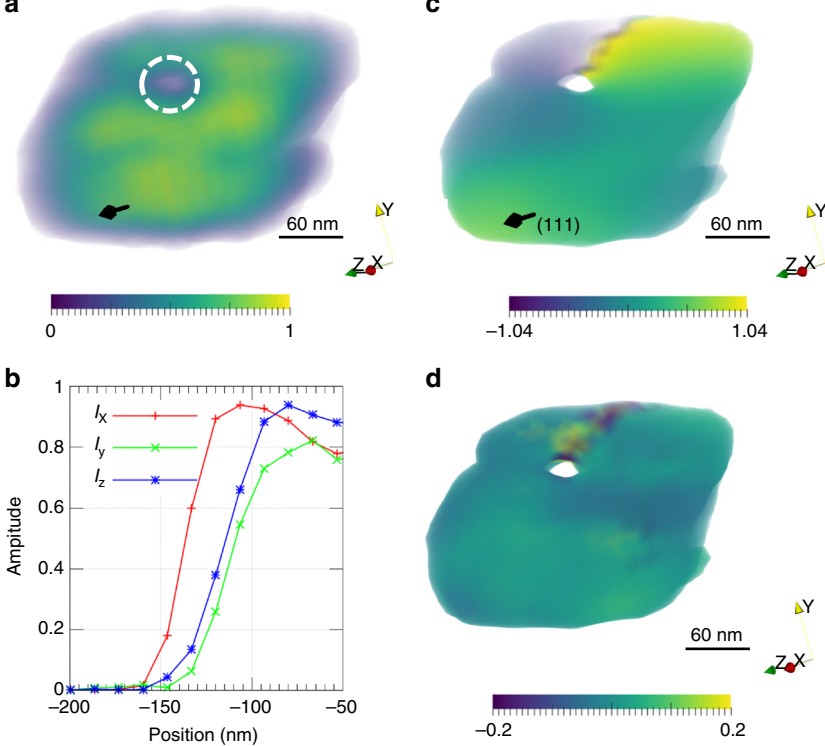

**Fig. 3** Reconstructed amplitude and strain in a Cu grain. **a** Reconstructed electron density showing a cylindrical region of low amplitude circled in white. Color is by amplitude. **b** Amplitude variation across the surface of the grain. **c**, **d** Volume renderings of displacement and strain in the grain projected along the (111) direction. **c** is colored by projected displacement in Å along the (111) direction while **d** is colored by the strain projected along the (111) direction

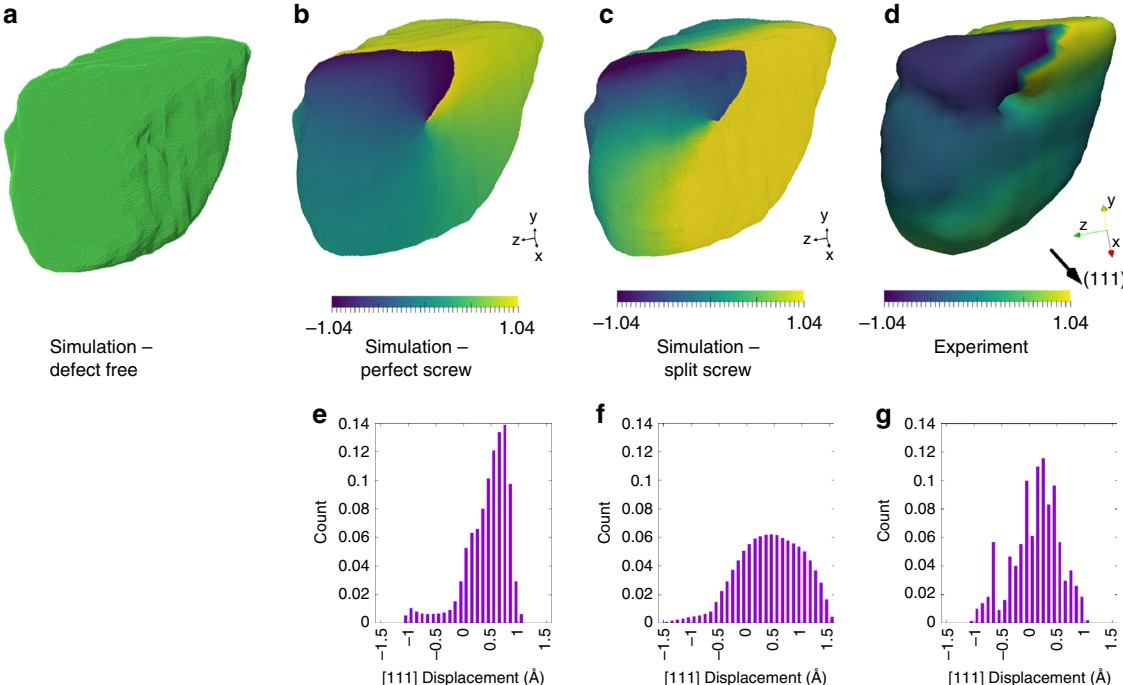

**Fig. 4** Atomic structure and displacement fields with and without a screw dislocation. **a** Atomistic structure obtained by filling Cu atoms into the experimentally observed grain. **b** Atomistic structure following addition of a perfect screw dislocation, and **c** atomistic structure after energy minimization. Atoms are colored by their displacement along the (111) from their initial structure. **d** isosurface rendering of the experimentally imaged Cu grain. **b**–**d** are colored by displacement projected along the (111) direction in Å. **e**–**g**, histograms of the displacement field corresponding to the images in **b**–**d**

to split into partials, even though such a state should be energetically favorable. We hypothesize that the constrains imposed by the adjoining grains, which are not imaged currently or included in the atomistic simulation might prevent the dissociation of the dislocation into partials. To test this hypothesis, we repeated the MD simulations keeping the atoms on the surface fixed. As expected, the screw dislocation did not split into partials for the same simulation conditions as described in Supplementary Fig. 1. Supplementary Fig. 2 shows that these results are independent of simulation size within what is computationally tractable. We note that while the analysis in this section suggests that a perfect screw dislocation might be present, our analysis is based on the measured displacement field only along one crystallographic orientation ({111}), and to make a definite judgment viz. perfect screw vs. partial screw vs. mixed dislocation, multiple Bragg directions will have to be measured and compared to theory. Supplementary Fig. 3 shows the result of an alternate analysis, where we imposed the experimentally observed displacement field (along {111}) on to the scaled down atomic model. Unsurprisingly, on energy minimization, the displaced atoms returned to the unstrained state except at the surface where the atoms were fixed. A different outcome might be expected if we could measure and impose the full strain tensor as the initial conditions to the atoms.

In conclusion, we used coherent X-ray diffraction to image the strain field within an individual grain of copper following the application of a tensile load. The imaging in conjunction with atomistic modeling revealed the presence of a screw dislocation within the grain. This study represents the first measurement of strain following mechanical loading in an unsupported polycrystalline film. Furthermore, for the first time, we have successfully developed a framework that enables complementary atomistic simulations to be performed on structure experimentally imaged through BCDI measurements. This integration between atomistic modeling and experiments can be leveraged to provide insight into dynamic processes that each alone cannot[16,36,37]. Such an approach has been demonstrated in integrating information from atom probe tomography and MD simulations to provide insight into dislocation networks in Ni-based superalloys[38]. Finally, in contrast to widely used strain measurement techniques such as digital image correlation and digital volume correlation that do not provide the resolution to resolve the strain field from individual dislocations, BCDI does not require the use of diffractive optics and as such, is only wavelength limited in resolution and is among the most photon-efficient of X-ray imaging techniques[39]. We foresee that such features of BCDI will prove to be particularly advantageous when attempting to resolve the structure and strain fields from more complex dislocation networks that arise during the work hardening of metals.

## Methods

**BCDI object reconstruction**. Real space images were reconstructed from the measured 3D coherent diffraction patterns using the guided optimization approach of Chen et al.[40] with ten individuals for four generations. The guided HIO approach performs phase retrieval on multiple (in this case ten) individuals simultaneously, each starting from a different random guess. At the end of each generation, the best individual (as quantified by the sharpness metric) is mixed with the other individuals. Phase retrieval is again performed using these individuals as starting guesses, and this process is repeated for four generations. The final image is obtained by averaging the three best individuals (as quantified by the error metric) from the last generation. Hence, each individual at each generation was the result of 620 iterations of hybrid input-output (HIO) and error reduction (ER)[41]. The measured diffraction data was significantly oversampled and so was binned by a factor of 2 to speed up the convergence of the iterative algorithms.

**MD simulation details**. Using the atom filled grain as the input structure, structural energy minimization was performed using the conjugate-gradient method as implemented in the LAMMPS package. The minimization was allowed to proceed until the relative change in energy between iterations was $<10^{-8}$, i.e., $dE/E<10^{-8}$. Shrink-wrap (non-periodic) boundary conditions were imposed in all directions and no constraints were imposed on any atoms in the grain. The embedded atom potential of Mishin et al.[42] was used, which was parameterized to accurately reproduce the stacking fault energy among other properties.

## Data availability

The experimental data, atomic structures, LAMMPS input files and Python code are available from MJC upon reasonable request.

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

## Acknowledgements

This work was supported by the U.S. DOE at LANL under Contract No. DE-AC52-06NA25396 through the LANL LDRD Program 20180683ER and the Institute for Materials Science 2016 Rapid Response program. This work was also supported by Argonne LDRD 2015-149-R1 (Integrated Imaging, Modeling, and Analysis of Ultrafast Energy Transport in Nanomaterials) and Argonne LDRD 2018-019-N0 (A.I. C.D.I: Atomistically Informed Coherent Diffraction Imaging). An award of computer time was provided by the Innovative and Novel Computational Impact on Theory and Experiment (INCITE) program. The experiment was performed at the 34-IDC beamline of the Advanced Photon Source (APS). This work was supported, in part, by the Center for Nanoscale Materials. The Advanced Photon Source, the Center for Nanoscale Materials and the Argonne Leadership Computing Facility are supported by the U.S Department of Energy, Office of Science, Office of Basic Energy Sciences, under Contract No. DE-AC02-06CH11357. The work was in part supported by the Center for Integrated Nano-technologies, a U.S. DOE BES user facility. We thank Dr. Richard Hoagland, Profs. Robert Suter and Anthony Rollett for several insightful discussions.

## Author contributions

S.J.F. and R.L.S. designed the research. M.J.C., R.P., T.S.O'.L., J.M., W.C., E.M., R.H., S.J.F. and R.L.S. performed the experiment. M.J.C. and T.S.O'.L. performed the data analysis and M.J.C. performed the atomistic simulations. J.K.B. synthesized the samples. All authors contributed to the discussion and writing of the manuscript.

## Additional information

**Competing interests:** The authors declare no competing interests.

