## [Peer Review File · Nature Communications]

Reviewers' comments:

Reviewer #1 (Remarks to the Author):

In its introduction, this manuscript by Cherukara et al promises really important results in materials science, measuring strains in a single grain of copper as a function of deformation. It falls short however in delivering this: there is no graph of any measured quantity changing as a function of strain. Indeed only single snapshots of grains are available, with no trends, apparently.

Even the "data" shown in Fig 4c and d only show two different models, and it is not clear which one agrees better with the data in 4e.

The ability to identify a single dislocation in the deformed material and show that it is not split into partials is an important result worthy of publication in Nature Comms, but not in the present form of the manuscript, which advertizes much more.

Minor points:

L101 by a coherent beam focused down to $\sim 700 \mu \times 700 \mu$, seems a bit large for nanoparticles. How was this determined?

L170 perfect screw dislocation using the conjugate-gradient method I think this should read "conjugate"

L194 Such seamless integration... this statement is unqualified. The connection between the diffraction measurement and the finite-element method is not described in the manuscript.

Reviewer #2 (Remarks to the Author):

Review of the article NCOMMS-17-14695

BCDI (Coherent diffraction Imaging in Bragg condition) has been demonstrated around 10 years ago. Since the pioneer results were dealing with model isolated sub-micronic objects (dewet Pb, Au or Cu island), the technique is now more mature and comes real engineering materials like polycrystalline thin films.

The article of Cherukara et al is coming in that new context. Tensile loading is a very useful mechanical test. However to make it applicable to BCDI is not easy. The authors proposed to use a free standing film (removal of MgO substrate after deposition) to perform the tensile loading. A numerical workflow is also developed to implement advanced simulation on a synthetic grain with an identical shape than the measured one (shape deduced from the retrieval procedure). This numerical study strengthens the phase reconstruction effectively. This numerical implementation is original and interesting.

However in both background and shape, the article presents some weaknesses. The paper is constructed as a first step through the characterization of "defect nucleation and dynamics" but at the end of the article the reader doesn't know if it will be possible with the described configuration (acquisition time, Bragg peak tracking during loading, etc...). I can understand that {in-situ tensile loading + 3D map acquisition + phase retrieval} remains difficult but after the abstract the reader is eager to see some dislocations "moving". Maybe abstract could be amended because in fine no in-situ analysis has been exposed in that paper. Moreover figures must be improved (missing scalebar, etc..) and the EBSD map is not commented in the text.

The work is going into the right direction and is answering well-founded questions. I believe this paper should be worth for publication in Nat. Comm if relevant corrections and precisions are offered in the second version.

Detailed Remarks

I have some concerns concerning the figures. Many technical information are missing.

#1 : line 987-993. Information about film disposition are exposed but I do not see the thickness of the film?

#2 : a nice EBSD map is shown in Fig. 2, but I find few comments in the text. Acquisition before or after the tensile loading? Do you have any chance to localize the diffracting grain on the EBSD map?

#3 : Fig. 4 e : no scalebar on the experimental data. This information (+ acquisition time) is essential to evaluate if in-situ testing will be possible effectively.

It seems that q resolution (or oversampling) is better for 4.e than in the simulation 4c,d. Has the $I_{\text{exp}}(q)$ been binned prior to the reconstruction? why?

An inset with line cuts could improve the visibility.

#4 : line 101 : The beam is "big" as compare to the size of the grain. Many photons are wasted? What is the dynamic in you $I(q)$ map? What is the recording time for the whole 3D map? Could you comment about the wavefront? Is-it comparable with the phase shift coming from the displacement field of the dislocation?

#5 : Fig. 1,3,4 are slightly redundant (I suggest to keep 2 figures in the text and put on in the supplemental). I hardly see the size of the object (only one scale in Fig. 3d).

Reviewer #3 (Remarks to the Author):

The present paper aims to develop a methodology to image dislocations, particularly screw dislocations, using X-ray Bragg coherent diffraction imaging (BCDI). The overall idea is indeed innovative and builds on the works of A. Prakash, E. Bitzek et al. (2015), *Acta Materialia*, 92, and M. Moody, S. Ringer et al. (2014), *Nat. Comm.* 5, in trying to utilise the synergy between experiments and simulations to validate experimental observations. In the current work, the authors propose to use BCDI to obtain the electron density and strain fields in an individual grain, and subsequently use them as input in atomistic simulations with LAMMPS. By correlating the displacement fields in experiments and simulations, the authors conclude the defect to be a screw dislocation. Screw dislocations are perhaps the most difficult to image in a TEM. The methodology proposed in this work could hence be a value addition not only in terms of imaging screw dislocations, but also towards designing materials by engineering defects.

The paper, however, has several severe shortcomings, a few of which are listed below.

1. There is severe lack of details in the manuscript, on the methods used in the current work. In the experimental section, elementary details like the Bragg conditions/angle, equipment use, etc., which are necessary for the reproduction of the data are missing.

The major problem, however, is with the atomistic simulations. No details on the boundary conditions (BCs) used for the simulations of fig. 5 are provided. No details on the force/energy norm used for static relaxation have been provided. How large was the structure in terms of number of atoms? Was the structure relaxed to the desired force/energy norm, or did the simulation merely stop because of the total number of function calls being reached? This has tremendous implications for the result presented in the current work! (see points 3 ff. below).

2. The authors could explain in more detail as to why they conclude that the defect is indeed a line

defect and e.g. not a void? Looking at fig. 3a and b, the region of low amplitude intensity is roughly 20nm. This seems to be rather large for a dislocation core itself. The authors must have their reasons for making this conclusion, and it would be helpful for the reader to clearly explain the same. Perhaps one could calculate the Eshelby twist that accompanies a screw dislocation instead of merely using the displacement?

3. Due to the absence of information on the boundary conditions in the atomistic simulations, one can only assume what could possibly be the BCs used in the current work. Assuming the two extremes of free BCs and fixed BCs for the surface atoms of the grain, one would still have to call the results presented in fig. 5 into question.

In the case of free boundary conditions, it is strange that the dislocation is held inside the grain. The lowest energy state would be that the dislocation would try to move to the surface and result in a dislocation free grain (there is no driving force for the dislocation to remain the grain, and the free boundary conditions must attract the dislocation towards itself). In the case of free BCs, the dislocation might first even cross slip onto a different plane.

In the case of fixed BCs, what was the resulting stress state in the grain? Since the dislocation does not even move from its original position, there must be stresses acting on it (assuming that the sample has been properly relaxed) and holding it in place! How high are these?

4. Why was the atomistic sample scaled down by a factor of 5? Was it merely to keep simulation times to a minimum? (I cannot approximate the computation times required, given the lack of number of atoms). How do the relaxed configuration differ with changing scaling factors? Has the reproducibility of the results been confirmed?

5. The results of the static relaxation in atomistic simulations (fig. 5) are strange. Copper is a material with low SFE (approx. 45 mJ/m²). The potential used in the current work also provides a similar value for the stable SFE (44.4 mJ/m²). As a result, I cannot see why a dislocation would not split into partials when it has been completely relaxed (provided, of course, that it does not move to a free surface!).

In view of this, I can only conclude that the system has not been sufficiently relaxed. I would suggest that the authors look very carefully at their relaxation methodology, and perhaps use a different tool (apart from LAMMPS) with a better relaxator to check their results.

6. The final statements in "Discussion" section of the paper are hence incorrect. Particularly the last statement

"We hypothesize that the constrains [sic] imposed by the adjoining grains, which are not included in the atomistic simulation prevent the dissociation of the dislocation into partials."

is troublesome. If the constraints of neighbouring grains are absent in the current simulations, then the dislocation must split into partials. There exists no reason for it to not do so! Comparing the case of no-constraints to a constrained grain in the polycrystalline material is essentially incorrect.

7. Comparing color maps is perhaps ok for a first approximation. To make quantitative and assertive statements, it is important to compare distributions. The authors must provide distributions of the displacement field in experiments and simulations. Even a voxelized approach for experiments would provide a decent distribution of the displacement field, which can then be used to compare with the atomic displacement field from simulations. Furthermore, where color maps need to be used, I would suggest that the authors use another colouring scheme like viridis, or simply a blue to red colouring scheme which makes things easy to compare. The rainbow colouring scheme, although pretty, is known to lead to misinterpretation of results.

8. Another drawback is that the current manuscript fails to put the work in context. How is the methodology presented here better than the work of Jesse Clark et al. (group of Ian Robinson from Oxford, and Ref. 3 in the current manuscript)? In other words what is the value addition with the new methodology proposed. This is particularly important since it involves a further step of

atomistic simulations, the nuances of which might not be known to the experimentalist. How is the methodology proposed different to what was proposed by Prakash et al. (2015), *Acta Materialia*, 92 (Group of Erik Bitzek, Erlangen)? A further piece of work which must be mentioned here is that of Yang et al. (2015), *Nature Communications*, 6 (group of Peter Nellist, Oxford) who used aberration corrected STEM to image screw dislocations.

In summary, although the presented idea is indeed enterprising, the lack of certain key details and the incorrect relaxation of the atomistic samples leads me to reject the paper in its current form.

Reviewers' comments:

Reviewer #1 (Remarks to the Author):

Reviewer comment 1:

In its introduction, this manuscript by Cherukara et al promises really important results in materials science, measuring strains in a single grain of copper as a function of deformation. It falls short however in delivering this: there is no graph of any measured quantity changing as a function of strain. Indeed only single snapshots of grains are available, with no trends, apparently.

Even the "data" shown in Fig 4c and d only show two different models, and it is not clear which one agrees better with the data in 4e.

The ability to identify a single dislocation in the deformed material and show that it is not split into partials is an important result worthy of publication in Nature Comms, but not in the present form of the manuscript, which advertises much more.

Author response:

We have modified the introduction and discussion to more accurately reflect the fact that we have recorded only the final state and not the evolving structure. We appreciate the statement from the reviewer that this work is an important result.

Changes in manuscript:

Edited abstract and introduction to reflect the changes suggested by the referee.

Reviewer comment 2:

Minor points:

L101 by a coherent beam focused down to $\sim 700 \mu \times 700 \mu$, seems a bit large for nanoparticles. How was this determined?

Author response:

We thank the reviewer for pointing this out, the units should read nm not microns. This has been corrected in the revised manuscript.

Changes in manuscript:

"...were illuminated by a coherent beam focused down to $\sim 700 \text{ nm} \times 700 \text{ nm}$, with the detector..."

Reviewer comment 3:

L170 perfect screw dislocation using the conjugant-gradient method I think this should read "conjugate"

Author response:

We thank the reviewer for pointing this out, and we have corrected this in the revised manuscript.

Changes in manuscript:

“...with a perfect screw dislocation using the conjugate-gradient method as implemented...”

Reviewer comment 4:

L194 Such seamless integration... this statement is unqualified. The connection between the diffraction measurement and the finite-element method is not described in the manuscript.

Author response:

We have not employed finite element simulations in this work. We have however utilized molecular statics simulations that were performed using the reconstructed grain structure. To the best of our knowledge, this is the first report where atoms have been filled into the reconstructed image from a CDI measurement and then subsequently used in an atomistic simulation.

Changes in manuscript:

“Furthermore, for the first time, we have successfully developed a framework that enables complementary atomistic simulations to be performed on the experimentally imaged structure. This integration between atomistic modeling and experiments can be leveraged to provide insight into dynamic processes that each alone cannot.”

Reviewer #2 (Remarks to the Author):

Review of the article NCOMMS-17-14695

Reviewer comment:

BCDI (Coherent diffraction Imaging in Bragg condition) has been demonstrated around 10 years ago. Since the pioneer results was dealing with model isolated sub-micronic objects (dewetted Pb, Au or Cu island), the technique is now more mature and comes real engineering materials like polycrystalline thin films.

The article of Cherukara et al is coming in that new context. Tensile loading is a very useful mechanical test. However to make it applicable to BCDI is not easy. The authors proposed to use a free-standing film (removal of MgO substrate after deposition) to perform the tensile loading. A numerical workflow is also developed to implement advanced simulation on a synthetic grain with an identical shape than the measured one (shape deduced from the retrieval procedure). This numerical study strengthen the phase reconstruction effectively. This numerical implementation is original and interesting.

However in both background and shape, the article presents some weaknesses. The paper is constructed as a first step through the characterization of ‘‘defect nucleation and dynamics’’ but at the end of the article the reader don’t know if it will be possible with the described configuration (acquisition time, Bragg peak tracking during loading, etc...). I can understand that {in-situ tensile loading + 3D map acquisition + phase retrieval} remains difficult but after the abstract the reader is eager to see some dislocations ‘‘moving’’. Maybe abstract could be amended because in fine no in-situ analysis has been exposed in that paper. Moreover figures must be improved (missing scalebar, etc..) and the EBSD map is not commented in the text. The work is going into the right direction and is answering well-founded questions. I believe this paper should be worth for publication in Nat. Comm if relevant corrections and precisions are offered in the second version.

Author response:

We thank the referee for this comment, we have modified the abstract and introduction to clarify that the imaging study followed tensile loading of the sample.

Changes in manuscript:

Edited abstract and introduction to reflect the changes suggested by the referee.

Reviewer comment:

Detailed Remarks

I have some concerns concerning the figures. Many technical information are missing.

#1 : line 987-93. Information about film disposition are exposed but I do not see the thickness of the film?

Author response:

We thank the reviewer for this question, the film is ~2 microns thick and we have included this in the manuscript.

Changes in manuscript:

“The Cu thin films which range from 1-2 microns in thickness were subsequently separated from the MgO substrate through sonication.”

Reviewer comment:

#2 : a nice EBSD map is shown in Fig. 2, but I find few comments in the text. Acquisition before or after the tensile loading? Do you have any chance to localize the diffracting grain on the EBSD map?

Author response:

We thank the reviewer for this question. The EBSD measurement was taken before the tensile loading. We did not attempt to locate the grain after the CXDI measurement. However, we plan to do this in our future studies using indents or other fiducials.

Changes in manuscript:

“Figure 2 shows the grain orientation and size distributions of an as-grown sample before tensile loading. Pole figures in Figure 2B show the crystallographic textures. From the pole figures, we can see that the majority of the grains with their crystal plane normal are oriented along the (111) direction, yielding a moderate texture of ≈ 4 multiples of random density.”

Reviewer comment:

#3 : Fig. 4 e : no scalebar on the experimental data. This information (+ acquisition time) is essential to evaluate if in-situ testing will be possible effectively.

It seems that q resolution (or oversampling) is better for 4.e than in the simulation 4c,d. Has the $I_{\text{exp}}(q)$ been binned prior to the reconstruction? why?

An inset with line cuts could improve the visibility.

Author response:

We thank the reviewer for these extremely insightful questions.

The experimental data set was binned 2x2 before running the iterative phase retrieval algorithm. Binning reduces the array size and consequently speeds up the phase retrieval considerably. Care was taken to ensure that the data set remained oversampled by a factor of 2 or more.

Changes in manuscript:

We have added a note on the binning of the data to the Methods section.

Reviewer comment:

#4 : line 101 : The beam is “big” as compare to the size of the grain. Many photons are wasted? What is the dynamic in you $I(q)$ map? What is the recording time for the whole 3D map? Could you comment about the wavefront? Is-it comparable with the phase shift coming from the

displacement field of the dislocation?

Author response:

We thank the reviewer for pointing this out, the beam size should have read 700nmX700nm. This has been corrected in the revised version of the manuscript. Each scan rocking curve scan is usually measured in approx. 20 mins. Any small incoherence in the incoming wavefront is corrected for by the use of partial coherence correction during the phase retrieval as described by Clark et al. (Clark, J. N., et al. "High-resolution three-dimensional partially coherent diffraction imaging." *Nature communications* 3 (2012): 993.)

Changes in manuscript:

Page 5:".. which is ideal for the X-ray spot size of 700 nm X 700 nm...."

Reviewer comment:

#5 : Fig. 1,3,4 are slightly redundant (I suggest to keep 2 figures in the text and put on in the supplemental). I hardly see the size of the object (only one scale in Fig. 3d).

Author response:

We thank the reviewer for this suggestion, and have merged Figs. 4 and 5 and worked to increase the readability of the scale bars in the figures.

Changes in manuscript:

New figure 3 and figure 4.

Reviewer #3 (Remarks to the Author):

The present paper aims to develop a methodology to image dislocations, particularly screw dislocations, using X-ray Bragg coherent diffraction imaging (BCDI). The overall idea is indeed innovative and builds on the works of A. Prakash, E. Bitzek et al. (2015), *Acta Materialia*, 92, and M. Moody, S. Ringer et al. (2014), *Nat. Comm.* 5, in trying to utilise the synergy between experiments and simulations to validate experimental observations. In the current work, the authors propose to use BCDI to obtain the electron density and strain fields in an individual grain, and subsequently use them as input in atomistic simulations with LAMMPS. By correlating the displacement fields in experiments and simulations, the authors conclude the defect to be a screw dislocation. Screw dislocations are perhaps the most difficult to image in a TEM. The methodology proposed in this work could hence be a value addition not only in terms of imaging screw dislocations, but also towards designing materials by engineering defects.

The paper, however, has several severe shortcomings, a few of which are listed below.

Author response:

We thank the reviewer for their positive comments on the value of this work and challenges associated with imaging screw dislocations. We have worked to address the shortcomings

outlined by the reviewer and list them below.

Reviewer comment:

1. There is severe lack of details in the manuscript, on the methods used in the current work. In the experimental section, elementary details like the Bragg conditions/angle, equipment use, etc., which are necessary for the reproduction of the data are missing.

The major problem, however, is with the atomistic simulations. No details on the boundary conditions (BCs) used for the simulations of fig. 5 are provided. No details on the force/energy norm used for static relaxation have been provided. How large was the structure in terms of number of atoms? Was the structure relaxed to the desired force/energy norm, or did the simulation merely stop because of the total number of function calls being reached? This has tremendous implications for the result presented in the current work! (see points 3 ff. below).

Author response:

We have added additional details on the BCDI experiment that are included in the section “Polycrystalline BCDI experiment”

Shrinkwrap (non-periodic) boundary conditions were used in the experiment. The structure consisted of ~5.5 million atoms and the minimization was stopped at the point the energy tolerance was met ($dE/E < 10^{-8}$).

Changes in manuscript:

We have included this information in the revised version of the manuscript under the Methods section.

Reviewer comment:

2. The authors could explain in more detail as to why they conclude that the defect is indeed a line defect and e.g. not a void? Looking at fig. 3a and b, the region of low amplitude intensity is roughly 20nm. This seems to be rather large for a dislocation core itself. The authors must have their reasons for making this conclusion, and it would be helpful for the reader to clearly explain the same. Perhaps one could calculate the Eshelby twist that accompanies a screw dislocation instead of merely using the displacement?

Author response:

We thank the referee for this insightful question. The large size of the void corresponding to the dislocation core is a consequence of the low resolution of the reconstructed images (30-50 nm). The region of low amplitude in conjunction with a displacement field that closely matches the field from the atomistic model, leads us to believe that the observed defect is indeed a screw dislocation.

Changes in manuscript:

Page 7: "... between the calculated atomic displacements as shown in Fig. 4 B and the experimentally observed lattice displacements shown in Fig. 4 D suggests that the experimentally observed phase structure is a result of the strain field from a screw dislocation.^{34,35} We note that the volume of low amplitude around the dislocation core appears

larger than expected (~20 nm), but is expected given the resolution of the reconstruction is (30-50 nm). Additionally, we compute the diffraction patterns from the atomistic structures with and without a screw dislocation as shown in Fig. 4 E-G. In..."

Reviewer comment:

3. Due to the absence of information on the boundary conditions in the atomistic simulations, one can only assume what could possibly be the BCs used in the current work. Assuming the two extremes of free BCs and fixed BCs for the surface atoms of the grain, one would still have to call the results presented in fig. 5 into question.

In the case of free boundary conditions, it is strange that the dislocation is held inside the grain. The lowest energy state would be that the dislocation would try to move to the surface and result in a dislocation free grain (there is no driving force for the dislocation to remain the grain, and the free boundary conditions must attract the dislocation towards itself). In the case of free BCs, the dislocation might first even cross slip onto a different plane.

In the case of fixed BCs, what was the resulting stress state in the grain? Since the dislocation does not even move from its original position, there must be stresses acting on it (assuming that the sample has been properly relaxed) and holding it in place! How high are these?

Author response:

We appreciate the insightful discussion of the reviewer on the boundary conditions and the static nature of the screw dislocation. We have worked to clarify the manuscript and address these concerns in the main text and supplemental materials. If the surface atoms were not fixed in the simulation, then indeed the dislocation does escape to the surface of the grain if minimized further. Consequently, we stopped the minimization at point that the dislocation completes splitting into partials. We have additionally performed the energy minimization with the surface atoms fixed as suggested by the referee (see responses below and in the supplementary materials).

Changes in manuscript:

Added details to the methods section. Supplementary figures comparing the effect of fixed and free boundary conditions.

Reviewer comment:

4. Why was the atomistic sample scaled down by a factor of 5? Was it merely to keep simulation times to a minimum? (I cannot approximate the computation times required, given the lack of number of atoms). How do the relaxed configuration differ with changing scaling factors? Has the reproducibility of the results been confirmed?

Author response:

Yes, the structure was scaled down by a factor of 5 to reduce the computational cost. The scaled structure has ~5.5 million atoms. We have also included Supplementary figure 2 showing the negligible effect of different scaling factors.

Changes in manuscript:

Page 6: "... The resulting volume was scaled down by a factor of 5 in every dimension and filled with FCC Cu atoms so that the (111) crystallographic direction is oriented along the

experimentally measured Q vector. The resulting structure has 5521267 atoms. We used the nanoSCULPT code...”

Added supplementary figure 2

Reviewer comment:

5. The results of the static relaxation in atomistic simulations (fig. 5) are strange. Copper is a material with low SFE (approx. 45 mJ/m²). The potential used in the current work also provides a similar value for the stable SFE (44.4 mJ/m²). As a result, I cannot see why a dislocation would not split into partials when it has been completely relaxed (provided, of course, that it does not move to a free surface!).

In view of this, I can only conclude that the system has not been sufficiently relaxed. I would suggest that the authors look very carefully at their relaxation methodology, and perhaps use a different tool (apart from LAMMPS) with a better relaxator to check their results.

Author response:

We thank the reviewer for identifying this very interesting aspect of the current work. Upon further analysis, the screw dislocation in the atomic model **does** indeed split into partials during energy minimization in the presence of free boundary conditions. However, as we have shown in the supplemental simulations and figures, for fixed boundary conditions the dislocations does not split into partials or move to the surface. Fig. 4 C shows the relaxed atomic structure colored by the displacement field projected along (111). The computed displacement field differs from the experimentally observed displacement field, which matches the displacement field from a pure screw dislocation. Indeed the dislocation does escape to the surface of the grain if minimized further. We stopped the minimization at point that the dislocation completes splitting into partials and begins to move to the closest free surface.

Changes in manuscript:

Page 9:” crystallographic direction before energy minimization (Fig. 4 A,B)) and following minimization (Fig. 4 C)). We note that further energy minimization to a higher tolerance in forces and energies leads to the dislocation escaping to the free surface of the grain. As is evident..”

Reviewer comment:

6. The final statements in “Discussion” section of the paper are hence incorrect. Particularly the last statement

“We hypothesize that the constrains [sic] imposed by the adjoining grains, which are not included in the atomistic simulation prevent the dissociation of the dislocation into partials.” is troublesome. If the constraints of neighboring grains are absent in the current simulations, then the dislocation must split into partials. There exists no reason for it to not do so! Comparing the case of no-constraints to a constrained grain in the polycrystalline material is essentially incorrect.

Author response:

As discussed earlier the dislocation does indeed split into partials in the unconstrained case. This has been clarified in the manuscript and figures. To simulate the constraints imposed by the neighboring grains, we repeated the simulation while keeping surface atoms fixed. As we have shown in the supplemental work, the dislocation does not move when surface atoms are fixed.

Changes in manuscript:

Reworked Figure 4 and accompanying text. Added supplementary figure 1.

Reviewer comment:

7. Comparing color maps is perhaps ok for a first approximation. To make quantitative and assertive statements, it is important to compare distributions. The authors must provide distributions of the displacement field in experiments and simulations. Even a voxelized approach for experiments would provide a decent distribution of the displacement field, which can then be used to compare with the atomic displacement field from simulations. Furthermore, where color maps need to be used, I would suggest that the authors use another colouring scheme like viridis, or simply a blue to red colouring scheme which makes things easy to compare. The rainbow colouring scheme, although pretty, is known to lead to misinterpretation of results.

Author response:

We thank the referee for these suggestions. We have changed all colour maps to viridis as suggested by the referee. We have also included distributions of the displacement field in Figure 4.

Changes in manuscript:

Figure 3 and 4 changed to the colour scheme suggested by the referee. Figure 4 now also includes histograms of the displacement field.

Reviewer comment:

8. Another drawback is that the current manuscript fails to put the work in context. How is the methodology presented here better than the work of Jesse Clark et al. (group of Ian Robinson from Oxford, and Ref. 3 in the current manuscript)? In other words what is the value addition with the new methodology proposed. This is particularly important since it involves a further step of atomistic simulations, the nuances of which might not be known to the experimentalist. How is the methodology proposed different to what was proposed by Prakash et al. (2015), *Acta Materialia*, 92 (Group of Erik Bitzek, Erlangen)? A further piece of work which must be mentioned here is that of Yang et al. (2015), *Nature Communications*, 6 (group of Peter Nellist, Oxford) who used aberration corrected STEM to image screw dislocations.

In summary, although the presented idea is indeed enterprising, the lack of certain key details and the incorrect relaxation of the atomistic samples leads me to reject the paper in its current form.

Author response:

We thank the referee for pointing us to these references. We have extended the introduction to include these references in the paper.

The most significant advancement from all prior BCDI measurements including Clark et al. (Ref. 3 in paper) is that the measurements described in this work were from an *unsupported free-standing* polycrystalline film, as opposed to isolated nanoparticles or supported thin films. Consequently, this work represents the first step to being able to image plastic deformation *in-situ* and in 3D. Secondly, our workflow that incorporates a reconstructed image with an atomic model is the first such integration between BCDI and MD simulations and has broad ramifications for the integrated characterization of a variety of processes ranging from catalysis, heat and mass transport etc. The work of Prakash et al. is similar in that they integrate insights from experiment and simulation, but the characterization technique they employ is completely different.

Changes in manuscript:

Page 2: “information. Recently 3D electron imaging using high-angle annular dark-field scanning TEM (ADF-STEM) tomography has been used to image dislocations in 3D,¹⁰ and techniques have been developed to quantify strain fields from dislocations in 2D.¹¹ ADF-TEM has also been used to image screw-dislocations side-on.¹² Electron microscopy techniques that can simultaneously image structure and strain in 3D however, remains elusive. Furthermore, TEM requires sample thicknesses of ~100nm, the preparation of which can lead to the artificial strains due to processing of the sample...”

Page 9:... complementary atomistic simulations to be performed on the experimentally imaged structure. This integration between atomistic modeling and experiments can be leveraged to provide insight into dynamic processes that each alone cannot.^{16,38,39} Such an approach has been demonstrated in integrating information from atom probe tomography and MD simulations to provide insight into dislocation networks in Ni-based superalloys.⁴⁰ Finally,...

Reviewers' comments:

Reviewer #1 (Remarks to the Author):

I have read the authors response to my comments and the revised manuscript. I think the work is now ready for publication.

Reviewer #2 (Remarks to the Author):

Dislocation imaging is of primary importance in material science. And remain a challenge for experimentalists. The work of Cherukara et al is original and paves the way to a new technique to study defect behavior in "real material".

The corrections and additions that have been made are valuable. The article is clear and meaningful for the readers. Moreover Cherukara et al have precisely responded to my concerns (beam size, sample morphology, wavefront correction, etc ...).

In conclusion, I believe the article is now worth to be publish in Nature Com.

Reviewer #3 (Remarks to the Author):

Review comments to the paper entitled "Three dimensional X-Ray diffraction imaging of dislocations in polycrystalline metals under tensile testing" by Cherukara et al., submitted to Nature Communications.

This review is in response to comments of the authors following the first review.

The authors have indeed clarified quite a few important points in the revised manuscript. There are still 2 major points that need addressing:

1. The authors now show the distribution of displacements from simulations and experiment to justify their conclusion that the defect is indeed a screw dislocation. A detailed look into, however, shows clear differences between the experiment and the perfect screw, with the latter being the basis of the conclusion in the paper (the perfect screw from the simulations has a more mirrored log-normal distribution, whilst fig. 4G (experiment) shows a seemingly Gaussian distribution). How do the authors conclude on the defect being a perfect screw dislocation despite this difference in distribution of displacements? (I understand that the colorful pics do seem to be close to each other, in comparison to the split screw, but given the lack of any other defects studied (like even a 60° or 30° dislocation), I am not sure how the authors can be completely confident in it being a screw dislocation)

1a. I do have a suggestion to help the authors substantiate their conclusion (if it is indeed a screw dislocation). Instead of applying the displacement field of a perfect screw dislocation, why not apply the displacements obtained from the experiment (even if the experimental data corresponds to projected displacement along one direction) onto the atomistic sample and relax the structure? This should be pretty straightforward to accomplish in a relatively short span of time. The system should -- under appropriate boundary conditions -- then find the optimal defect structure itself!

2. The statement in the discussion section

Furthermore, for the first time, we have successfully developed a framework that enables complementary atomistic simulations to be performed on the experimentally imaged structure.

is incorrect. At best, this might be regarded true only for the BCDI data. That such complementary atomistic simulations can be performed on experimentally imaged data (like Atom probe data, SEM images, amongst others) has been shown previously in many works - a few of which have also been cited here.

Reviewers' comments:

Reviewer #1 (Remarks to the Author):

Reviewer comment:

I have read the authors response to my comments and the revised manuscript. I think the work is now ready for publication.

Author response:

We thank the referee for all of her/his suggestions during the review process and acknowledge their contribution to the improvement of the manuscript.

Reviewer #2 (Remarks to the Author):

Reviewer comment :

Dislocation imaging is of primary importance in material science. And remain a challenge for experimentalists. The work of Cherukara et al is original and paves the way to a new technique to study defect behavior in ‘‘real material’’. The corrections and additions that have been made are valuable. The article is clear and meaningful for the readers. Moreover Cherukara et al have precisely responded to my concerns (beam size, sample morphology, wavefront correction, etc ...). In conclusion, I believe the article is now worth to be publish in Nature Com.

Author response:

We thank the referee for all of her/his suggestions during the review process and acknowledge their contribution to the improvement of the manuscript.

Reviewer #3 (Remarks to the Author):

Review comments to the paper entitled ‘‘Three dimensional X-Ray diffraction imaging of dislocations in polycrystalline metals under tensile testing’’ by Cherukara et al., submitted to Nature Communications.

Reviewer comment:

This review is in response to comments of the authors following the first review.

The authors have indeed clarified quite a few important points in the revised manuscript. There are still 2 major points that need addressing:

1. The authors now show the distribution of displacements from simulations and experiment to justify their conclusion that the defect is indeed a screw dislocation. A detailed look into, however, shows clear differences between the experiment and the perfect screw, with the latter being the basis of the conclusion in the paper (the perfect screw from the simulations has a more mirrored log-normal distribution, whilst fig. 4G (experiment) shows a seemingly Gaussian distribution). How do the authors conclude on the defect being a perfect screw dislocation

despite this difference in distribution of displacements? (I understand that the colorful pics do seem to be close to each other, in comparison to the split screw, but given the lack of any other defects studied (like even a 60° or 30° dislocation), I am not sure how the authors can be completely confident in it being a screw dislocation)

1a. I do have a suggestion to help the authors substantiate their conclusion (if it is indeed a screw dislocation). Instead of applying the displacement field of a perfect screw dislocation, why not apply the displacements obtained from the experiment (even if the experimental data corresponds to projected displacement along one direction) onto the atomistic sample and relax the structure? This should be pretty straightforward to accomplish in a relatively short span of time. The system should -- under appropriate boundary conditions -- then find the optimal defect structure itself!

Author response:

We thank the referee for her/his insightful comment and careful reading of the manuscript and data. We attribute the difference between the experimental and model distributions of displacement to the fact that the model only includes displacement from the screw dislocation. In reality, the strain state of the grain is a result not only of the screw dislocation but also from additional elastic stresses imposed by neighboring grains.

We thank the referee for her/his clever suggestion of overlaying the observed displacement field onto the atom filled structure. Unfortunately, during relaxation, the displaced atoms relax to their unstrained state. As the referee points out, we only have knowledge of the displacement field in one direction and this is an insufficient constraint on the atoms. The energy minimization was performed as before (as described in **methods**) while keeping the surface atoms fixed.

A Experimental data

B Initial atomic structure

C Relaxed atomic structure

Figure 1: A Slice through the experimental image. B initial atomic displacement imposed along the Q direction. C relaxed atomic structure showing displacement along the Q. The only remaining displacement is seen on the surface where the atoms were kept fixed.

Finally, we have added a statement qualifying the uncertainty in characterizing the observed defect as a screw. We acknowledge that while the comparison to modeling is suggestive of the defect being a perfect screw, to make a definite judgement, we would need to measure the displacement along multiple Bragg orientations.

Changes in manuscript:

Page 10: “We note that while the analysis in this section suggests that a perfect screw dislocation might be present, our analysis is based on the measured displacement field only along one crystallographic orientation ($\{111\}$), and to make a definite judgement viz. perfect screw vs partial screw vs mixed dislocation, multiple Bragg directions will have to be measured and compared to theory.”

Reviewer comment :

2. The statement in the discussion section

Furthermore, for the first time, we have successfully developed a framework that enables complementary atomistic simulations to be performed on the experimentally imaged structure is incorrect. At best, this might be regarded true only for the BCDI data. That such complementary atomistic simulations can be performed on experimentally imaged data (like Atom probe data, SEM images, amongst others) has been shown previously in many works - a few of which have also been cited here.

Author response:

We thank the referee for highlighting this and also for pointing us to the relevant references in her/his previous comments to us. We have corrected the statement to be specific to BCDI measurements.

Changes in manuscript:

Page 10: “Furthermore, for the first time, we have successfully developed a framework that enables complementary atomistic simulations to be performed on structure experimentally imaged through BCDI measurements.”

REVIEWERS' COMMENTS:

Reviewer #3 (Remarks to the Author):

Many thanks to the authors for the changes to the manuscript.

I would like to clarify one particular thing in my last comment: When I suggested to impose the displacements from experiments onto the atomistic sample and relax the structure, I meant not a total relaxation, but merely a local relaxation in order to obtain the core structure of the dislocation correctly (cf. the work of Bitzek et al. [2009], MSMSE 17, pp 055008, where a local version of the FIRE algorithm is used to obtain the core structure of the dislocation loop). When a total relaxation is performed, as is the case here, it is unsurprising that the dislocation disappears, due to --as the authors point out-- the lack of the right boundary conditions (stresses imposed by neighborhood of the current grain). Consequently, the boundary conditions are more relevant and important than the measurement of the displacement field from multiple Bragg directions, since even the right displacement field measured via different directions, it cannot be guaranteed that the dislocations stays in place without the right boundary conditions.

I am, however, happy with the changes made to the manuscript. Nevertheless, I would suggest two very minor changes: include this new result as a part of supplementary material refer to the lack of appropriate boundary conditions in the simulations